# A consensus *S. cerevisiae* metabolic model Yeast8 and its ecosystem for comprehensively probing cellular metabolism

Hongzhong Lu[1,5], Feiran Li[1,5], Benjamín J. Sánchez[1], Zhengming Zhu[1,2], Gang Li [1], Iván Domenzain [1], Simonas Marcišauskas [1], Petre Mihail Anton [1], Dimitra Lappa [1], Christian Lieven [3], Moritz Emanuel Beber [3], Nikolaus Sonnenschein[3], Eduard J. Kerkhoven [1] & Jens Nielsen [1,3,4]

Genome-scale metabolic models (GEMs) represent extensive knowledgebases that provide a platform for model simulations and integrative analysis of omics data. This study introduces Yeast8 and an associated ecosystem of models that represent a comprehensive computational resource for performing simulations of the metabolism of *Saccharomyces cerevisiae*––an important model organism and widely used cell-factory. Yeast8 tracks community development with version control, setting a standard for how GEMs can be continuously updated in a simple and reproducible way. We use Yeast8 to develop the derived models panYeast8 and coreYeast8, which in turn enable the reconstruction of GEMs for 1,011 different yeast strains. Through integration with enzyme constraints (ecYeast8) and protein 3D structures (proYeast8[DB]), Yeast8 further facilitates the exploration of yeast metabolism at a multi-scale level, enabling prediction of how single nucleotide variations translate to phenotypic traits.

[1] Department of Biology and Biological Engineering, Chalmers University of Technology, Kemivägen 10, SE412 96 Gothenburg, Sweden. [2] School of Biotechnology, Jiangnan University, 1800 Lihu Road, 214122 Wuxi, Jiangsu, China. [3] The Novo Nordisk Foundation Center for Biosustainability, Technical University of Denmark, DK-2800 Kgs Lyngby, Denmark. [4] BioInnovation Institute, Ole Maaløes Vej 3, DK2200 Copenhagen N, Denmark. [5] These authors contributed equally: Hongzhong Lu, Feiran Li. Correspondence and requests for materials should be addressed to J.N. (email: nielsenj@chalmers.se)

I n the era of big data, computational models are instrumental for turning different sources of data into valuable knowledge for e.g. biomedical[1,2] or industrial use[3]. As a bottom-up systems biology tool, genome scale metabolic models (GEMs) connect genes, proteins and reactions, enabling metabolic and phenotypic predictions based on specified constraints[4,5]. The quality and scope of GEMs have improved as demonstrated by the models for human[6], yeast[7], and *E. coli*[8]. With their thousands of reactions, genes and proteins, GEMs also represent valuable organism-specific databases. Therefore, it is important to keep track of changes as new knowledge is added to GEMs, in order to make model development recorded, repeatable, free and open to the community. Approaches to record updates of a GEM exist for this purpose[9,10], albeit with some limitations on their simplicity and flexibility.

*S. cerevisiae* is a widely used cell factory[11,12] and is extensively used as a model organism in basic biological and medical research[13,14]. Recently, the emergence of technologies, such as CRISPR[15] and single cell omics data generation[16], have accelerated the developments in systems biology. Consistent with strong research interests in yeast, the relevant GEMs have also undergone numerous rounds of curation since the first published version in 2003[17]. These GEMs have contributed significantly to systems biology studies of yeast including their use as platforms for multi-omics integration[18,19], and use for in silico strain design[20,21]. However, the hitherto latest version, Yeast7[22], with only 909 genes, falling behind the latest genome annotation, presents a bottleneck for the use of yeast GEMs as a scaffold for integrating omics datasets.

Metabolism is complex and regulated at several different levels[23,24]. Traditional GEMs only consist of reactions and their related gene and protein identifiers, and therefore cannot accurately predict cellular phenotypes under varied environmental conditions other than nutritional conditions, e.g. simulating the impact of temperature on growth rates[25]. Recently, enzyme constraints[26] and protein 3D structures[27] have been integrated into GEMs, thereby expanding their scope of application and laying the foundation for whole cell modelling. GEMs constrained with $k_{cat}$ values and enzyme abundances have been able to directly integrate proteomics data and correctly predict cellular phenotypes under conditions of stress[26]. GEMs with protein 3D structures connect the structure-related parameters and genetic variation[27] with cellular metabolism[6], thus enlarging the prediction scope of GEMs. However, it remains challenging to directly predict cellular metabolism based on changes in protein sequences with current GEMs. Advanced functional mutation cluster analysis constrained with protein 3D structures is therefore needed to integrate knowledge on protein structures into GEMs.

This study presents Yeast8, the latest release of the consensus GEM of *S. cerevisiae*[22,28–30]. We also introduce a model ecosystem around this GEM, including ecYeast8, a model incorporating enzyme constraints; panYeast8 and coreYeast8, representing the pan and core metabolic networks of 1011 *S. cerevisiae* strains; and proYeast8[DB], a database containing 3D structures of metabolic proteins. This model ecosystem has the ability to meet wide application demands from the large scientific yeast community in systems and synthetic biology of yeast.

Yeast8 is a consensus GEM maintained in an open and version-controlled way. Through ecYeast8 and proYeast8[DB], multiple parameters related to protein kinetics and 3D structures could be integrated based on gene–protein-reaction relations. Furthermore, with panYeast8 and coreYeast8, 1011 strain-specific GEMs were reconstructed and compared. Thus, with Yeast8 and its model ecosystem, we demonstrate that the metabolism of yeast can be characterised and explored in a systematic way.

## Results

**Recording community developments of yeast GEMs with GitHub**. We devised a general pipeline to record updates to the model using Git (https://git-scm.com/), a version control system, and GitHub (https://github.com/), a hosting service for Git repositories (Fig. 1a and Supplementary Fig. 1). Hereby, we record everything related to updates of the GEM, including datasets, scripts, corrections and each released version of the GEM (Supplementary Fig. 1). This Git version-controlled model enables open and parallel collaboration for a wide community of scientists. With Git and GitHub, each version of the yeast GEM can be released periodically, which helps to promote the simultaneous development of a model ecosystem around yeast GEM (Fig. 1b).

**Increasing the scope of the yeast metabolic network**. We systematically improved the yeast GEM while moving from Yeast7 to Yeast8 through several rounds of updates (Fig. 1c and Supplementary Fig. 2). To improve the genome coverage, we added additional genes from iSce926[31]. Besides, all functional gene annotations of *S. cerevisiae* from SGD[32], BioCyc[33], Reactome[34], KEGG[35] and UniProt[36] were collected and compared (Supplementary Fig. 3) to update gene–protein-reaction relations (GPRs), as well as adding more GPRs. With Biolog experiments, i.e. evaluation of growth on a range of different carbon and nitrogen sources (Supplementary Data 1), and metabolomics mapping (see methods), extra reactions were added to enable the model to ensure growth on the related substrates, as well as connecting those metabolites with high confidence with the GEM. The biomass equation was modified by adding nine trace metal ions and eight cofactors. Additionally, 37 transport reactions were added in order to eliminate 45 dead-end metabolites. To improve lipid constraints, we reformulated reactions of lipid metabolism using the SLIMEr formalism, which Splits Lipids Into Measurable Entities[37]. As SLIMEr imposes additional constraints on both the lipid classes and the acyl chain distribution from metabolomics data, it improved the model performances in lipid metabolism.

In each round of model updates, standard quality-control tests, such as reaction mass balance check and ATP yield analysis, were performed. The results in Supplementary Fig. 2 and Supplementary Fig. 4 indicate that the gene (reaction) coverage in the model and its performance were improved during the iterative update process, which was also shown by comparing Yeast8 to Yeast7 (Fig. 1d–f). To facilitate the multi-omics integrative analysis and visualisation, we established a map of yeast metabolic pathways in SBGN (System Biology Graphical Notation) format (Supplementary Fig. 5) using CellDesigner[38].

**Expanding Yeast8 to enable enzyme constraints**. To enhance model prediction capabilities of Yeast8, ecYeast8 was generated by accounting for enzyme constraints using the GECKO framework[26] (Fig. 1b, Supplementary Table 1). In enzyme-constrained models, metabolic rates are constrained by the intracellular concentration of the corresponding enzyme multiplied by its turnover number ($k_{cat}$ value) to ensure that fluxes are kept at physiologically possible levels. Compared with Yeast8, ecYeast8 has a large reduction in flux variability for most of the metabolic reactions in both glucose-limited and glucose-excess conditions (Fig. 2a).

To further evaluate the predictive strength of ecYeast8, we compared the predicted maximum growth rate to that obtained from experiments in which we cultivated yeast (strain S288c) on 322 different combinations of carbon and nitrogen sources using microtiter plates (Fig. 2c). Here, the measured maximum specific growth rate for each carbon source under all nitrogen sources was

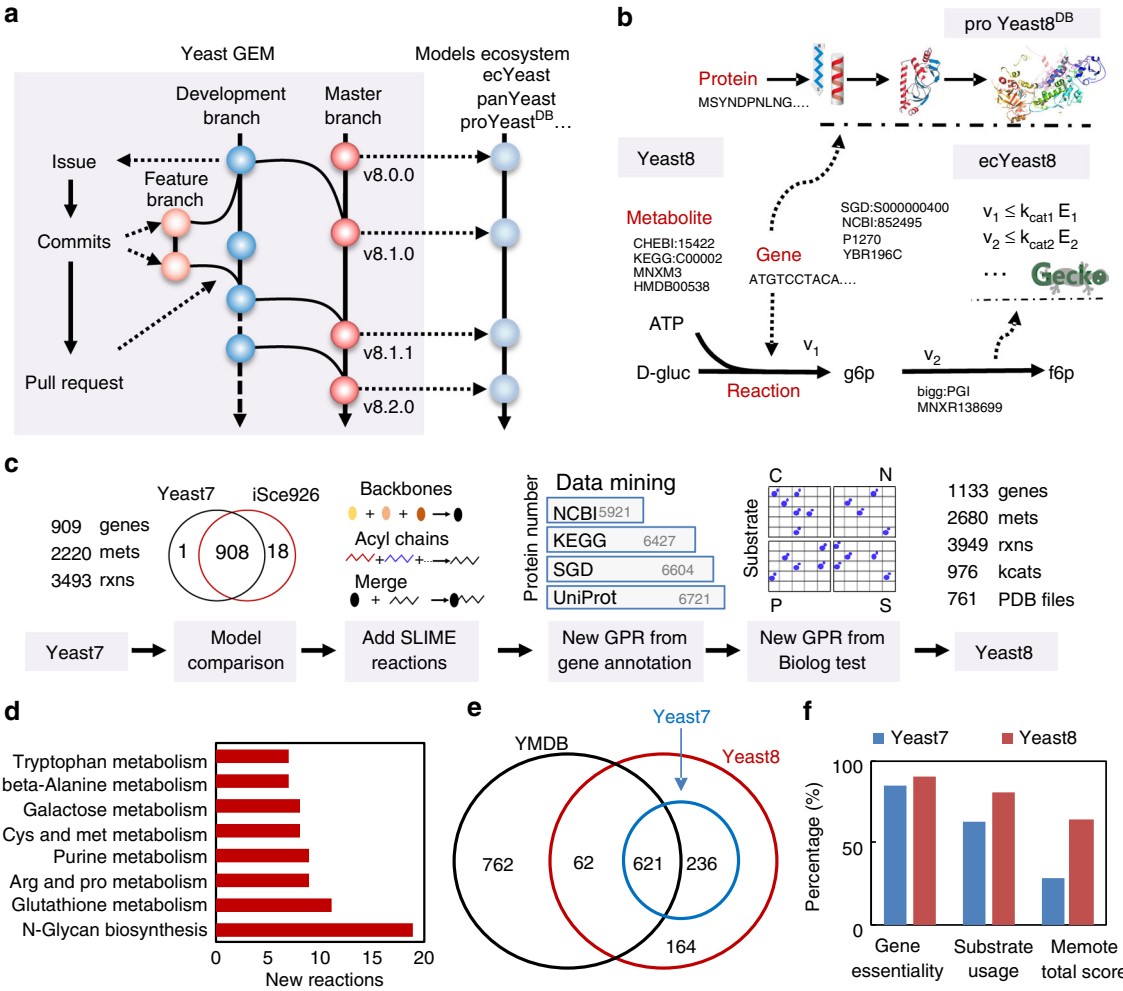

**Fig. 1** Framework of the yeast GEM project. **a** Recording model updates in a community way using GitHub. **b** In Yeast8, genes, metabolites and reactions are annotated with their corresponding IDs from different databases, which simplifies translation between namespaces. Yeast8 forms the basis of the model ecosystem from which proYeast8DB, ecYeast8, etc., are derived. **c** Major steps of development from Yeast7 to Yeast8. **d** Subsystem statistical analysis for the reactions added to Yeast8. **e** Metabolomics mapping between Yeast7, Yeast8 and the YMDB database. **f** Comparison of Yeast7 and Yeast8 in percent accuracy of gene essentiality and substrate usage analysis, as well as in memote test total scores (divided by 100)

scaled based on the growth on ammonia as a reference, with results displaying that the mean error for the predicted and scaled measured cell growth rate in ecYeast8 was 41.9%, which is significantly lower than using Yeast8 and traditional flux balance analysis alone (Fig. 2d–f). With ecYeast8, it is also possible to estimate flux control coefficients (FCC) of enzymes for growth on different carbon sources (Fig. 2b), i.e. enzymes exerting the majority of flux control on growth. For instance, growing on glucose, the major flux controlling enzyme is one of the isoforms of glyceraldehyde-3-phosphate dehydrogenase (Tdh1) whereas on fermentative carbon sources, such as ethanol and acetate, the majority of control is with Oli1, a key component of ATP synthase, which has earlier been identified to be important for respiratory metabolism in *S. cerevisiae*[39]. This type of analysis using ecYeast8 can provide clues for in silico cell factory design[26].

**Generation of panYeast8 and coreYeast8.** To investigate the correlation between phenotype and genotype, we used genomics data of 1011 *S. cerevisiae* strains from a recent genome-sequencing project and designed a pipeline to reconstruct GEMs for each strain (Supplementary Fig. 6a)[40]. Similar to the CarveMe method[41], a pan model for all yeast strains was firstly

reconstructed based on Yeast8 and the comprehensive pangen-ome annotation (Supplementary Note 1). Using panYeast8 and the gene presence matrix for the 1,011 strains[42], strain-specific GEMs (ssGEMs) could be generated automatically (Supplementary Fig. 6a). The reaction number in the models was found to be in range from 3969 to 4013 (Fig. 3a and Supplementary Fig. 7a).

After reconstruction of the ssGEMs, we formulated coreYeast8 based on shared reactions, metabolites and genes in all the strains, which contained 3895 reactions, 2666 metabolites and 892 genes. Even though there were 478 variable genes, only 147 reactions were found to be variable due to the existence of a large number of isoenzymes, which suggested that during the evolution of yeast, the increased number of gene copies with similar functions increased the robustness of cellular function at the protein level. We calculated the ratio of accessory genes (not included in the coreYeast8) based on subsystems defined in the KEGG database and found that the accessory genes were primarily engaged with alternative sugar metabolism and other secondary metabolism pathways (Supplementary Fig. 8). The number of reactions in coreYeast8 was relatively close to Yeast8, signifying that *S. cerevisiae* metabolism is well conserved among the 1011 yeast strains. It should be noted that coreYeast8 reflects the core metabolic functions for all *S. cerevisiae* strains, and can therefore

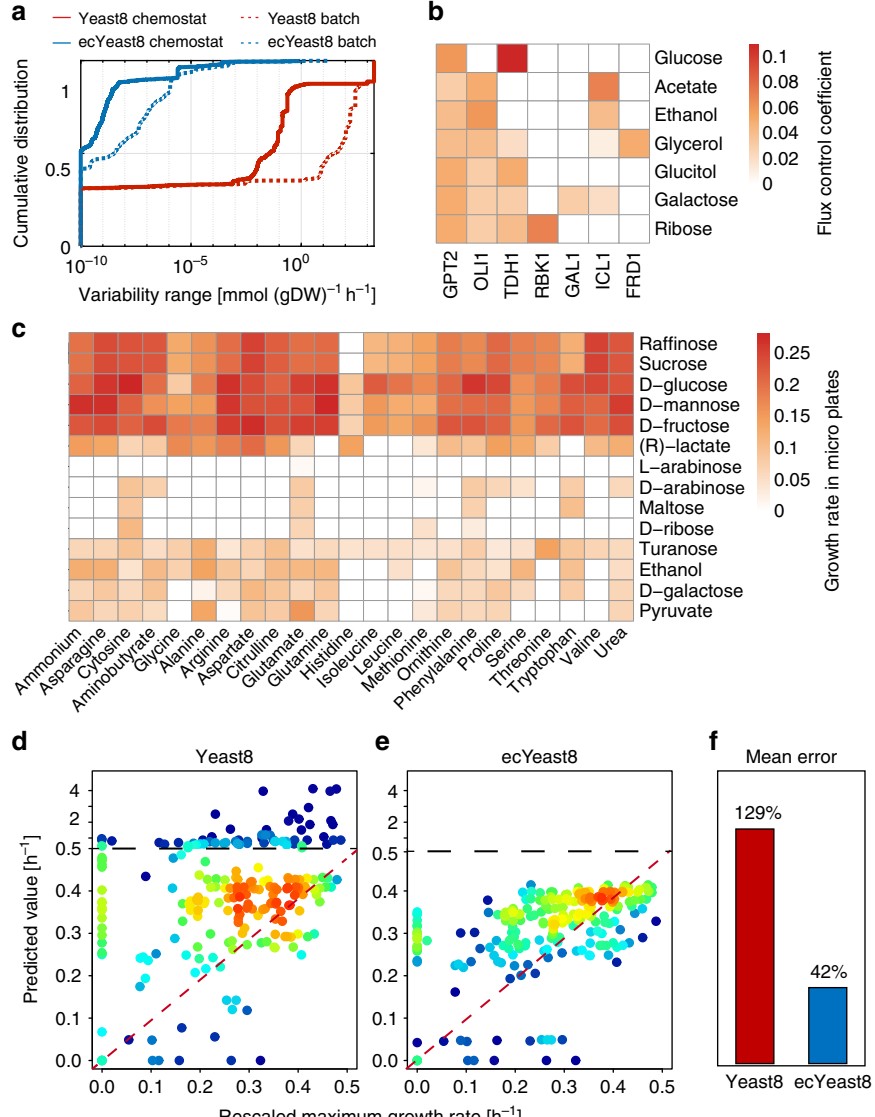

**Fig. 2** ecYeast8 with enzyme constraints shows improved fit over the regular Yeast8 metabolic model. **a** Comparison of model performance between ecYeast8 and Yeast8 based on flux variability analysis (FVA). **b** Flux control coefficient analysis with ecYeast8 simulated on different carbon sources in minimal medium with the growth as the objective function. **c** Growth rate of *S. cerevisiae* S288C in micro-plate under different combinations of carbon and nitrogen sources. **d**, **e** Prediction of maximum specific growth rates under different combinations of carbon and nitrogen sources using Yeast8 (**d**) and ecYeast8 (**e**), the red colour zones mean that the data points are overlapped due to higher density. **f** Mean errors for comparison of measured and predicted growth rates using Yeast and ecYeast8

be seen as a benchmark for reconstructing GEMs for other yeast strains. Similar to the core *E. coli* GEM[8], we observed that coreYeast8 could not grow in silico in a minimal medium since it lacks the ability to synthesise all required precursors for biomass production (Supplementary Note 1, Supplementary Table 2). This can be partly related to the definition of 'core biomass compositions' in our model, which still hints that the ability of particular yeast strains to grow on a minimal medium seems to be an acquired phenotype.

**Evaluation of 1011 strain-specific GEMs.** Using the 1011 GEMs, we estimated in silico substrate usage (including carbon, nitrogen and phosphorus) as well as the yield of 26 metabolites and biomass for growth on minimal media (Fig. 3b–f). From the substrate usage analysis, we found that some of the domesticated strains from the ecological origin "Industrial" had a smaller range

of substrates usage, suggesting that the domestication process may have resulted in the loss of functions that were not routinely used. As for the yield of biomass, it could be observed that the yield varied greatly between strains from different ecological origins. By comparison, the biomass yield was relatively lower for strains with the ecological origin 'Human', i.e. strains that have been isolated from humans (Fig. 3b), likely due to an adaptation towards growth on complex media, as many building blocks are provided by the host and the yeasts have adapted to these conditions.

To further compare the metabolic potential of different strains, we calculated the maximum yields of 20 amino acids and six key precursors in a minimal medium from 1011 ssGEMs (Fig. 3f) and the results illustrated that the gene background can have a remarkable effect on the ranges of maximum yields of desired products. Additionally, taken strains from 'Industrial' ecological origin as an example, a considerable variation in the strains'

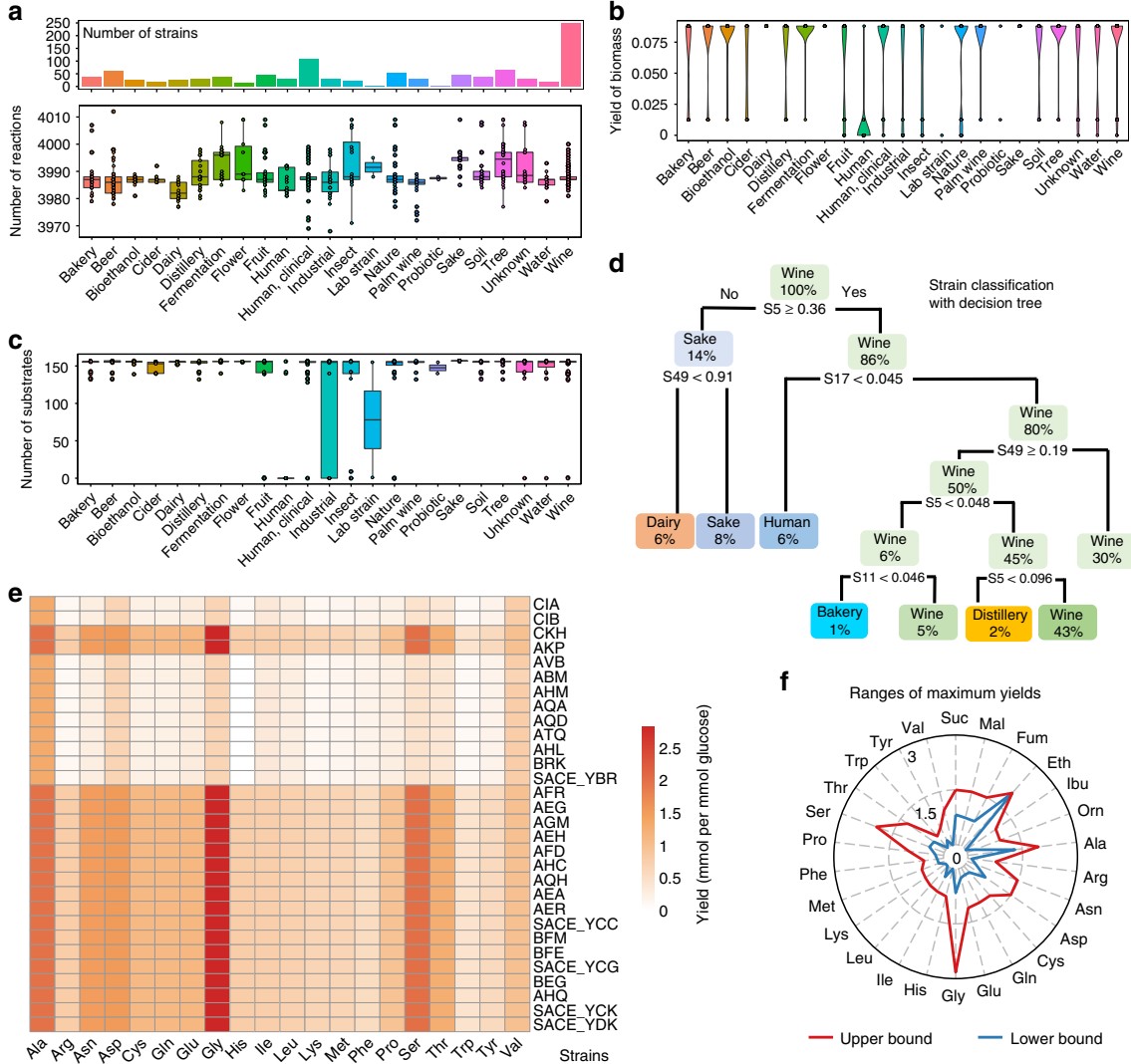

**Fig. 3** Conservation and diversity analysis of yeast metabolism through reconstruction of 1,011 strain-specific GEMs (ssGEM). **a** Amount of reactions of ssGEMs for strains from different ecological origins. The top column graph describes the strain number from each ecological origin. Different strains from one ecological origin could have the equal number of reactions. In the box plots of this work, if not mentioned, the bold line in the box represents the median. The lower and upper bounds of box show the first and third quartiles, respectively, and whiskers indicate ± 1.5× the interquartile range (IQR). The color dots overlaid represent the corresponding data points. **b** Prediction of biomass yield using ssGEMs on minimal media with glucose as the carbon source. **c** Number of substrates which can be used in silico for strains from different ecological origins. In the simulation using ssGEMs, 58 carbon sources, 46 nitrogen sources, 41 phosphate sources and 12 sulphate sources were used respectively in the minimal media. **d** Decision tree classification of strains according to the in silico maximum growth rate on different carbon sources (S5: Latic acid, S11: Serine, S17: Sorbitol, S49: Trehalose). The in silico uptake rate for each carbon source was set at 10 mmol (gBiomass)$^{-1}$ h$^{-1}$. **e** Comparison of the yield of 20 amino acids from glucose for 30 strains from the group of "industrial strain". **f** Comparison of in silico range of maximum yields of 20 amino acids and six key chemicals for all 1011 ssGEMs

capability to synthesise these amino acids was found (Fig. 3e). Combining genotype information, model simulation and phenotype data[40] revealed that variations in the simulated maximum yields of amino acids are primarily due to differences in the energy pathways used for ATP regeneration and the absence of some essential genes in amino acid synthesis (Supplementary Note 1). This shows that simulations with ssGEMs can be instrumental to evaluate the potential of a strain in producing specific chemicals, as well as enable analysis of the relation between genotype and phenotype.

Next, we classified the strains based on the genes and reactions contained in the 1,011 ssGEMs, as well as the in silico substrate usage. Using PCA analysis we found that the strains could be subdivided into several distinct groups according to the existence of genes and reactions (Supplementary Fig. 7e, f). However,

strains from different ecological origins can be clustered together, reflecting the high conservation in yeast metabolism. A decision tree algorithm was further employed to classify strains based on their substrate usage, while this initially indicated that strains from 'Human' can be separated from 'Wine' only based on the maximum growth rate on two substrates (lactic acid and sorbitol, Fig. 3d). When using hierarchical cluster analysis based on the reaction existence in ssGEMs for yeast strains from specific ecological origins, taking 'Human' as an example, these strains could be further divided into smaller groups based on the variable reactions shared between these strains (Supplementary Fig. 7g). The above analysis makes it clear that the model itself and the related simulation can classify different yeast strains, which in turn can be complementary to classical strain classifications based on single-nucleotide polymorphism (SNP) data[40].

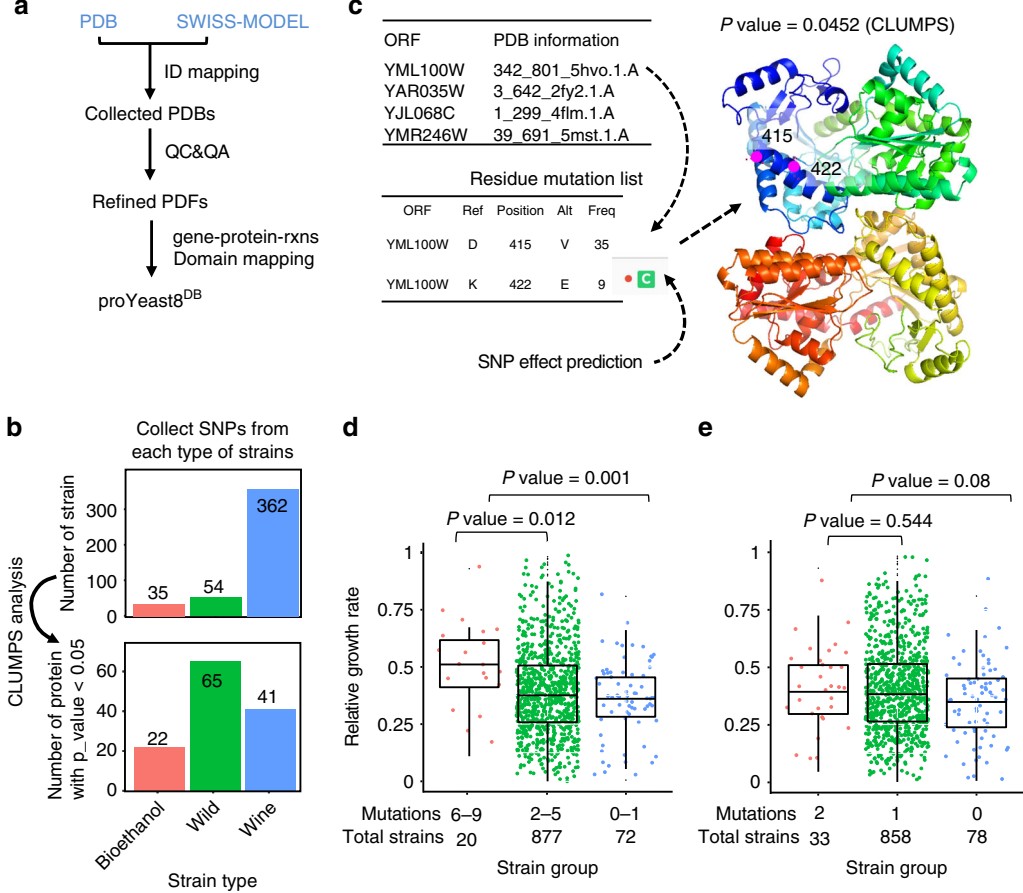

**Fig. 4** proYeast8[DB]—connects protein 3D structures with yeast GEM and enables the SNP mapping analysis based on protein structures. **a** Overview of proYeast8[DB] formulation. **b** CLUMPS analysis to find proteins with significant mutation enrichment[45]. **c** Function annotation of residue mutation from YML100W and mapping those mutations onto the protein structure. The 'C' means the related residue mutation is located on the conservative zone of proteins based on the function annotation. **d** Classification of strains based on residue mutations in proteins encoded by YML100W, YAR035W, YJL068C and YMR246W. **e** Classification of strains based on residue mutations from only one protein encoded by YML100W. The statistical analysis is based on Wilcoxon rank sum test

**Characteristics of metabolism displayed with proYeast8[DB].** Phenotype is not only determined by absence or presence of genes, but also by SNPs. As SNPs occur randomly, we need a way to test which SNPs are significant, by looking at their distribution over the protein structure. To enable such analysis, we collected PDB files for all the metabolic proteins in Yeast8, generating the proYeast8[DB] database (Supplementary Fig. 9a and Fig. 4a). Low quality PDB files were filtered out in order to only maintain high quality PDB files for our following analysis (Supplementary Note 2, Supplementary Fig. 9b–e).

To explore the potential functions of protein residue mutations, all the 16,258,509 homozygous SNPs from the 1011 *S. cerevisiae* genome-sequencing project[40] were collected (Supplementary Fig. 10a, b) and classified as synonymous SNPs (sSNPs) and nonsynonymous SNPs (nsSNPs). The distribution and correlation of relative values in nsSNPs and sSNPs (Supplementary Fig. 10c, d) stated clearly that there were fewer nsSNPs than sSNPs, meaning that the variation in protein sequences was significantly smaller than variations in nucleotide sequences. An analysis concerning the correlation between nsSNPs in genes with the corresponding protein abundances and number of reactions catalyzed by the corresponding protein was further conducted (Supplementary Fig. 10e, f). These results hinted that genes having high protein abundance and encoding enzymes catalyzing a large number of reactions tended to have fewer nsSNPs. From

this, it transpires that genes catalyzing abundant enzymes or enzymes with many metabolic functions are likely to be more conserved in evolution. In addition, enzymes with high flux control coefficients related to the growth on different carbon sources (Fig. 2d) had fewer nsSNPs (Supplementary Fig. 11), which shows genes with a high impact on cell growth may be more conserved[43].

To get further insight into which parts of metabolism may be sensitive or prone to mutations we chose the top 30 genes with the smallest and largest number of relative nsSNPs and conducted a GO-term analysis using DAVID 6.7[44] (Supplementary Table 3 and 4). Genes with the least amount of nsSNPs were enriched in thiamin metabolic process (*P* value = 0.0001) and glycolysis (*P* value = 0.0003), indicating that these pathways are conserved in the evolution of *S. cerevisiae*. By comparison, genes with the most nsSNPs were enriched in organic acid biosynthetic process (*P* value = 0.02), carboxylic acid biosynthetic process (*P* value = 0.02), etc.

Following the findings mentioned above, we then mapped all nsSNPs onto the 3D protein structures of proYeast8[DB] using CLUMPS method[45], which could find proteins with significant SNP clusters within the protein structures. As an example, using the above nsSNP data as input, we ran the CLUMPS pipeline for three different groups of *S. cerevisiae* strains: Wild, Wine, and Bioethanol strains (Fig. 4b). With a cut-off *P* value of 0.05,

proteins with significant mutation enrichment from each group of strains were identified (Fig. 4b). From the Wine group, we found GO terms like 'electron transport chain' ($P$ value = 0.018) and 'glutathione metabolic process' ($P$ value = 0.006, Supplementary Table 5), with glutathione metabolism being known to be distinct for wine yeasts, where it plays a role in anti-oxidation of various aromatic compounds[46,47].

Interestingly, we discovered that the Bioethanol group contained four proteins (YML100W, YAR035W, YJL068C, YMR246W) belonging to the GO term 'cellular response to heat' ($P$ value = 0.041, Supplementary Table 6), among which, YML100W is a large subunit of the trehalose 6-phosphate synthase/phosphatase complex, which is known to contribute to survival from acute heat stress. Two mutations were identified in the protein encoded by YML100W, both are located in a small cluster in the protein's 3D structure, despite being separated by seven residues in the primary protein sequence (Fig. 4c). The annotation from the mutfunc database[48] (http://mutfunc.com/) suggested that mutation at site of 422 in YML100W occurred in a conserved zone (Fig. 4c). We further wanted to know whether the mutations in these four proteins affected the phenotype of $S.$ $cerevisiae$ strains. Firstly, we classified all the strains based on residue mutations from these four proteins (Fig. 4e). It could be found that the relative growth rate at 42 °C was significantly higher for strains with more than six mutations in these four proteins than for strains that had less than two mutations. Similarly, the relative growth rates based on mutations only from YML100W were also analyzed, which indicated that the contribution of the two mutations from this single protein is smaller than over six mutations from all four proteins (Fig. 4f), further demonstrating that the ability to grow at elevated temperatures requires epistatic interactions of mutations in several genes (or proteins).

**Systematic analysis of ecYeast8 and proYeast8$^{DB}$.** To further display the value of the yeast model ecosystem, we combined ecYeast8 and proYeast8$^{DB}$ (Fig. 5a) to identify genes and their mutations that are related to a specific phenotype. Based on growth rate data provided in Peter et al.[40] (Fig. 5b), we selected 50 strains with the highest, medium, and lowest relative growth rate on complex medium with glycerol as the carbon source (growth with glucose was used as reference) (Fig. 5c). We then ran the hotspot analysis pipeline based on proYeast8$^{DB}$ to find hotspot zones of proteins for each group of strains. In parallel, we performed in silico flux control analysis to calculate FCCs for each protein by using ecYeast8 with growth as the objective function on the same medium. Hotspot analysis demonstrated that each group of strains has a set of unique hotspot zones in different proteins (Fig. 5d).

With the FCCs as filtering criteria, four proteins were identified with relatively high FCC (>0.01) from 50 strains with highest relative growth rate, among which YJL052W, one of the isozymes of glyceraldehyde-3-phosphate dehydrogenase (GAPDH), is located in the key pathway related to glycerol utilisation (Fig. 5f). When we listed all mutations of YJL052W, there exist six residue mutations at site of 31, 73, 24, 70, 125 and 248. Interestingly, we found four mutations at site of 24, 31, 70 and 73 were close to each other, far away from the other two mutations at site of 125 and 248 (Fig. 5e). The mutation at position of 31 and 73 were identified successfully using our hotspot analysis pipeline and the remaining two mutations at position of 24 and 70 could be filtered out according to the definition of important clusters in a hotspot zone, which should be made up of the significant pairs of two residues, separated by at least 20 amino acids in the original protein sequence[49]. When the CLUMPS analysis was employed

for the different combinations of mutations in the above site, the $P$ value for mutations at site of 24, 31, 70 and 73 is 0.0247 and $P$ value for mutations at site of 31 and 73 is 0.0195 (Supplementary Table 7). However, if all the six mutated positions are considered, the $P$ value is 0.7817. Therefore, mutations at position of 24, 31, 70 and 73 as a whole could form into a larger hotspot zone. From the protein functional annotation[36], we found that this hotspot zone was very close to the binding site of NAD$^+$ at site of 33 and 78.

We queried whether there were correlations between mutations occurring in hotspot zones and the growth phenotype. From the remaining 746 strains which were not used for the above hotspot analysis, we found eight strains that exhibit more than one mutation in the hotspot zone and another 15 strains having two mutations in YJL052W, but in the non-hotspot zone. Comparing the relative growth rate of these two groups of strains with the remaining, we saw that the former 8 strains grows faster on complex medium with glycerol as the carbon source than both the latter 15 strains, as well as all other strains not containing the mutations in hotspot zone (Fig. 5g), indicating that the mutation that happened in the hotspot zone is possibly beneficial for cell growth. Therefore, such an comprehensive analysis with our model ecosystem can enable identification of the potential mutation targets related to a specific phenotype.

## Discussion

As part of this study, we have developed Yeast8 aided by version control and open collaboration, which has provided a platform for a continued community-driven expansion of the model. This platform can greatly accelerate iterative updates of the model, and we believe that this approach should become the future standard for developing GEMs for other organisms. Yeast8 is the currently most comprehensive reconstruction of yeast metabolism, but it also represents a model that can be used for simulations. The platform provided through the GitHub repository enables addition of new knowledge when it is acquired as well as using this for further improving the model for simulations. Yeast8 is in line with the latest trend of performing model quality-control analysis in a standardised manner with memote[50]. Integrating consistent model evaluation with community model development will be instrumental to accelerate high quality development of GEMs.

Through developing Yeast8, we have significantly improved the metabolic scope of the consensus GEM of $S.$ $cerevisiae$. As more evidences from experiments and bioinformatics analyses are revealed and utilised to update GEMs, these models will move closer to the in vivo network. Based on Yeast8, we developed strain specific GEMs, enzyme-constrained GEMs (ecYeast8), etc., which together form a model ecosystem around the yeast GEM and improve cellular phenotype predictions. As an example, ecYeast8 verifies that the yeast phenotypes are to a large extent determined by protein resources allocation, which is consistent with recent research[39]. We expect predictions of ecYeast8 to further improve as more organism-specific kinetic data becomes available, hopefully generated in a high-throughput and systematic way[51]. By comparing panYeast8, coreYeast8 and 1,011 strain specific models, it can be concluded that metabolic capabilities are largely conserved for all $S.$ $cerevisiae$ strains, which is consistent with a recent study[52]. However, through strain specific GEM simulations, we have found subtle metabolic differences among the strains in the utilisation of substrates and the maximum yield of 26 chemicals. Exploring these differences constrained with more physiological data can guide future metabolic engineering and help to evaluate the potential of any given strain for any desired product, as well as provide clues about the mechanisms of evolutionary adaption. Currently, only

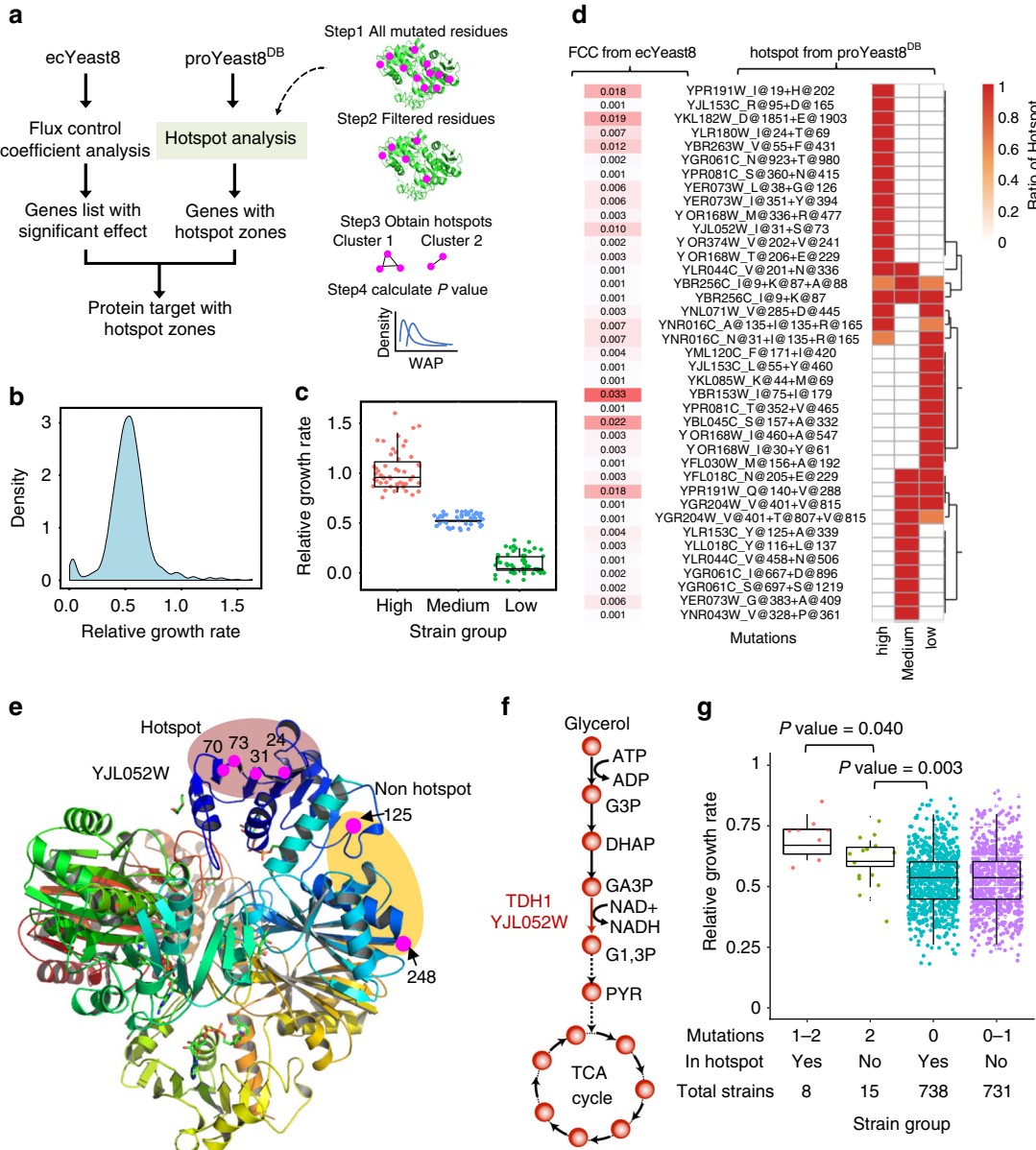

**Fig. 5** Systematic analysis of ecYeast8 and proYeast8[DB] underlies the potential relations between gene mutations and growth phenotype. **a** Pipeline to conduct the integrative analysis of ecYeast8 and proYeast8[DB]. **b** Distribution of relative growth rate with glycerol as carbon substrate in complex medium. **c** Three groups of strains with high, medium and low relative growth rates used for hotspot analysis. **d** Heatmap of hotspot zones for three groups of strains and heatmap of flux control coefficients obtained from ecYeast8 with growth as the objective function. **e** Mapping of all YJL052W mutations from strains with high growth rate onto the protein 3D structure. **f** Position of enzyme encoded by YJL052W in the glycerol utilisation pathway. **g** Classification of the remaining strains based on mutations in hotspot zones and non-hotspot zones of YJL052W. The statistical analysis is based on Wilcoxon rank sum test

in silico simulations were conducted using 1011 yeast ssGEMs; therefore, future experimental evidence for other non-reference strains will be important to evaluate the reliability of our model predictions.

With the increasing use of adaptive laboratory evolution in metabolic engineering it is important to have methods for rapid assessment of mutations in strains with improved phenotypes. Our multi-scale model analysis will be easily applicable in this area. As a further extension of Yeast8, we developed proYeast8[DB] by collecting and evaluating the yeast metabolic protein 3D structures from public databases[53,54]. This enabled identification of mutational hotspots associated with specific phenotypes. Furthermore, through combining the predicted targets from ecYeast8 and proYeast8[DB], we demonstrated how to identify mutations in enzymes with high flux control over a given pathway, which may

also be associated with desirable phenotypes. It should be noted that although proYeast8[DB] is useful for connecting GEMs to protein structure information like PDB identifiers and protein parameters, it is still challenging to directly predict phenotypes using the model with protein structure variations as input. High quality 3D protein structures at genome scale are still scarce[53], thus the breakthrough in 3D proteins structure simulations[55] and the related residues functional predictions are strongly expected. Nevertheless, like Recon3D[6] and the *E. coli* GEM-PRO[27], the proYeast8[DB] holds value as a means to explore the relation between the cell genotype and phenotype with clear evidence.

Yeast8 will continue to be developed together with its ecosystem of models. As a whole, they are expected be a solid basis for developing a whole cell model of an eukaryal cell, which may serve as a stepping stone to a wider use of model simulations in

life sciences, resulting in reducing the costs of developing biotechnology processes and drug discovery.

## Methods

**Tracking model changes with version control.** Git and GitHub were used to develop yeast-GEM in a traceable way. Git is used to track any changes of yeast-GEM, which are stored online in a GitHub repository (Supplementary Fig. 1). The structure of the yeast-GEM repository on GitHub contains the following three main directories:

(1) ComplementaryData, which contains the related database annotation and physiological data used for yeast-GEM updates. This data is generally stored as tab-separated value (.tsv) format for easier tracking of changes; (2) ComplementaryScripts, which contains all the scripts used to update yeast-GEM; (3) ModelFiles, which contains different formats of yeast-GEM for various applications. The.txt and.yml (YAML) formats make it convenient to visualise any changes in GitHub or Git clients. The.xml (SBML) format makes it easy to import the model across different toolboxes and programming languages.

As a standard step, a commit is needed when updating yeast-GEM. To make commits easy to understand, semantic commit messages are used (Supplementary Fig. 1c). To enable parallel model development, different branches of yeast-GEM are used, including a 'master' branch and a 'devel' (development) branch. Developers, and even other people from the community, can create new branches from the development branch to introduce their changes, and then request to merge them back through pull-requests. These changes are only merged to the development branch, and in turn the changes in the development branch are merged periodically to the master branch, which contains the stable releases of the model.

**General procedures used to standardise annotation of metabolites and reactions.** For the newly added reactions, their MetaNetX IDs were obtained according to a direct search in the MetaNetX[56] database using the related metabolite name or EC number information. MetaNetX IDs were also obtained by reaction ID mapping from the KEGG[35], Rhea[57] and BioCyc[33] databases. The reaction reversibility was corrected based on the BioCyc and BiGG databases[58]. MetaNetX IDs were also used to obtain the EC number for the corresponding reactions. As the MetaNetX database does not have the reaction name information, the name of each new reaction was obtained based on the reaction ID mapping in databases of KEGG, ModelSeed and BioCyc.

The compartment annotation of new reactions was refined based on information from the UniProt[36] and SGD[32] databases. The subsystem annotation was firstly obtained from KEGG[35], and if no subsystems were found there, information from BioCyc or Reactome[34] was used instead. If the reaction had no gene relations, we assumed that it occurred in the cytoplasm.

For all the metabolites contained in newly added reactions, the related MetaNetX IDs were obtained based on the reaction MetaNetX IDs. If not available, they were obtained by ID mapping based on KEGG IDs or ChEBI IDs. Once the metabolite MetaNetX IDs were obtained, the charge, formula, KEGG IDs and ChEBI IDs were obtained for the correspondent metabolite based on metabolites annotation in MetaNetX.

**Model update from Yeast7 to Yeast8.** Firstly, all the annotations regarding metabolite ChEBI IDs and KEGG IDs (Supplementary Table 8) were corrected in the latest version of the consensus GEM of yeast (version 7.6) based on the metabolite annotation available in KEGG and ChEBI[59]. Additionally, several genes from iSce926[31] that were not included in yeast 7.6 were added, as with all genes related to metabolic processes and transport in SGD, BioCyc, Reactome, KEGG and UniProt. The main databases used for model curation could be found in Supplementary Table 9.

In the Biolog experiments, the strain S288c was grown on 190 carbon sources, 95 nitrogen sources, 59 phosphorus sources, and 35 sulphur sources. The result showed that S288c could grow on 28 carbon sources, 44 nitrogen sources, 48 phosphorus sources and 19 sulphur sources. Based on these results new essential reactions were added to make the model capable of predicting growth on the related substrates. Meanwhile, all the metabolomics data contained in the YMDB database (measured metabolites) and the latest metabolomics research (Supplementary Table 10) were collected and compared with that in yeast GEM. A standard annotation was given for all these metabolites and a pipeline was designed to add the metabolites into the GEM without bringing any new dead-end metabolites. Detailed procedures in model curation are available in the Supplementary Methods.

**Model validation with varied experimental data sources.** To compare the metabolites coverage, the YMDB database[60] was parsed. There are 2024 metabolites for yeast, among which 871 were measured in *S. cerevisiae*. For each metabolite, ChEBI ID and KEGG ID were assigned, and based on them the corresponding MetaNetX ID was matched. For metabolites from Yeast7 and Yeast8, the MetaNetX ID of each metabolite was also obtained based on ID mapping.

The in silico growth on 190 carbon sources, 95 nitrogen sources, 59 phosphorus sources, and 35 sulphur sources were calculated and compared with phenotype data. The results can be divided into a confusion matrix, which contains: (1) G/G: in vivo growth/in silico growth (true positive); (2) NG/NG: in vivo no growth/in silico no growth (true negative); (3) G/NG: in vivo growth/in silico no growth (false negative); (4) NG/G: in vivo no growth/in silico growth (false positive);

The model quality is then evaluated based on accuracy (Eq. 1) and the Matthews' Correlation Coefficient (MCC)[61] (Eq. 2). Accuracy ranges from 0 (worst accuracy) to 1 (best accuracy). MCC ranges from $-1$ (total disagreement between prediction and observation) to $+1$ (perfect prediction).

$$\text{Accuracy} = \frac{\text{TP} + \text{TN}}{\text{TP} + \text{TN} + \text{FT} + \text{FN}} \quad (1)$$

$$\text{MCC} = \frac{\text{TP} \times \text{TN} - \text{FP} \times \text{FN}}{\sqrt{(\text{TP} + \text{FP})(\text{TP} + \text{FN})(\text{TN} + \text{FP})(\text{TN} + \text{FN})}} \quad (2)$$

To conduct gene essentiality analysis, we used the essential gene list from the Yeast Deletion Project, available at http://www-sequence.stanford.edu/group/yeast_deletion_project/downloads.html, which was generated from experiments using a complete medium. Accuracy and MCC were computed as described above.

The simulated aerobic and anaerobic growth under glucose-limited and nitrogen-limited conditions were compared with reference data[62]. The following procedure was employed to simulate chemostat growth in glucose-limited conditions. Firstly set the lower bound of glucose and $O_2$ uptake reactions using experimental values. Glucose and oxygen uptake fluxes are negative and therefore the lower bounds are fixed to represent the maximum uptake rates. Secondly maximise the growth rate.

As for nitrogen-limited conditions, since protein content in biomass drops dramatically under nitrogen-limited conditions, the biomass composition was rescaled according to reference conditions[63], then set the lower bound as measured for $NH_3$ and $O_2$ uptake reactions using experimental values and finally maximise the growth rate.

**Visualisation of Yeast8.** The maps of yeast-GEM were drawn for each subsystem using cellDesigner 4.4[38] (Supplementary Fig. 5). In-house R scripts were used to produce the map of each subsystem automatically based on Yeast8. Afterwards, the graph layout was adjusted manually in cellDesigner 4.4 to improve its quality and the whole yeast map in SBGN format could be found in https://github.com/SysBioChalmers/Yeast-maps/tree/master/SBMLfiles.

**Generation of ecYeast8.** The ecYeast8 model was generated based on the latest release of the GECKO toolbox, available at https://github.com/SysBioChalmers/GECKO. For each reaction, the algorithm queries all the necessary $k_{cat}$ values from the BRENDA database[64], according to gene annotation and a hierarchical set of criteria, giving priority to substrate and organism specificity. The $k_{cat}$ values are then added to reactions according to:

$$-\frac{1}{k_{cat}^{ij}} v_j + e_i = 0 \quad (3)$$

$$0 \leq e_i \leq [E_i] \quad (4)$$

$$v_j \leq k_{cat}^{ij} \cdot [E_i] \quad (5)$$

where $v_j$ represents the flux through reaction j, $e_i$ represents the amount of enzyme allocated for reaction j, $E_i$ represents the total concentration of enzyme i, and $k_{cat}$ represents the highest turnover number available for enzyme i and reaction j. The detailed procedure to generate ecYeast8 can be found in the supplementary material of the GECKO paper[26].

**Simulations with ecYeast8.** To predict the maximum growth rate under different carbon and nitrogen sources using ecYeast8, the following procedure was used. Firstly remove any constraints for the related uptake rates of carbon and nitrogen sources. Next, set minimal media made up of the related carbon and nitrogen sources. Lastly, simulate a growth rate maximisation, whereby the optimal value is fixed for posterior minimisation of the total protein usage. This provides a parsimonious flux distribution.

For comparative FVA between Yeast8 and an ecYeast8, the maximum growth rate and the optimal glucose uptake rates obtained with ecYeast8 are used as fixed value and upper bound, respectively, in the original GEM in order to perform a fair comparison of flux variability for the same growth phenotype.

Flux control coefficients (FCCs) are defined as a ratio between a relative change in the flux of interest and a relative change in the correspondent $k_{cat}$ of 0.1%, which can be described by:

$$\text{FCC}_i = \left(\frac{v_{up} - v_b}{v_b}\right) / \left(\frac{1.001 k_{cat}^{ij} - k_{cat}^{ij}}{k_{cat}^{ij}}\right) \quad (6)$$

where $v_b$ and $v_{up}$ are the original flux and new fluxes respectively when the $k_{cat}$ is increased by 0.1%.

**Re-annotation of the pan-genome from the 1011 yeast genome-sequencing project.** To construct the pan model of yeast (panYeast8), the latest genomics research by Peter et al has consulted[40]. In Peter's study, 1011 yeast strains genomes had been sequenced and analysed. A pan-genome was obtained from all these strains, made up of by 6081 non-redundant ORFs from *S. cerevisiae* S288C reference genome, and 1715 non-reference ORFs (nrORFs) from the other strains. For the 7796 ORFs, a panID was given for each of them. By comparison, 4940 ORFs are conserved in all these strains while 2846 ORFs are variables across all these strains. The annotation of non-redundant 6081 ORFs can be taken directly from the latest *S. cerevisiae* S288C genome annotation, while related gene–protein-reactions (GPR) can be obtained from Yeast8 directly.

As mentioned in Peter's article there are 774 nrORFs with the ortholog genes from *S. cerevisiae* S288C genome[40]. The blast analysis, along with the gene annotation of KEGG web service[35], and EggNOG web service[65], were employed to check and improve the original ortholog relation. To evaluate the ortholog gene relations qualitatively, the bi-directional blast hit (BBH) analysis was further conducted using Diamond[66]. Here the best hit in BBH analysis with pidentity larger than 80% were finally chosen and prepared for a panYeast8 formulation.

To further search reliable new reactions connected with nrORFs, the annotation results from KEGG and the EggNOG web service were used. According to the format request for the two web services, the protein fasta files of pan-genome were uploaded onto KEGG (https://www.genome.jp/tools/kaas/) and EggNOG (http://eggnogdb.embl.de/#/app/emapper). For the KEGG annotation, a BBH (bi-directional best hit) assignment method with the default parameters was used. For the EggNOG annotation, the HMMER with the default parameters was used. In the EggNOG annotation, each protein will be mapped onto KO ID and BiGG reaction ID while for the KEGG annotation, each protein will be given a unique KO ID. So if the KO ID for a protein is different between KEGG and EggNOG, then the KO ID given by KEGG will be preferred in the further analysis. If the KO ID was given for one protein by EggNOG, but not in KEGG, then this annotation will be also used for the pan-genome annotation. When the KO ids are obtained, the lists of KOs from nrORFs are compared with the reference ORFs. New KO ids for the nrORFs were subsequently extracted. Following this, the rxnID was obtained based on KO-rxnID mapping from KEGG database.

**Generation of panYeast8, coreYeast8 and strain specific GEMs.** For ortholog genes (e.g. gene C) obtained from pan-genome annotation, they can be merged based on the reference gene (e.g. gene A) function in the original model according to the following rules: (1) if A or B catalyze the same isoenzyme, the GPR rule could be changed to 'A or B or C' in panYeast8; (2) if A and B belong to a complex, the GPR rule should be updated from 'A and B' into '(A and B) or (C and B)'. Secondly, 51 new reactions with 13 new genes were merged into panYeast8. As for the genes identity in the model, in order to reduce chaos, the original gene IDs and gene names from original Yeast8 were kept, while for newly added genes, the panIDs defined in Peter's work[9] were used to represent the gene name.

Collapsed genes in pan-genome could be found in yeast GEM, and will be replaced with the corresponding ortholog genes defined in pan-genome. ssGEMs for 1011 strains were reconstructed based on panYeast8 along with the related strains specific genes list (Supplementary Fig. 6a). A Matlab function was developed to generate strain specific models automatically. Based on current gene existence information, if one gene from a complex is missing, then the reaction is removed; and if a gene from two isoenzymes is missing, then the reaction will be kept, though the GPRs will be updated to remove the missing gene. After the reconstruction of 1011 ssGEMs, coreYeast8 was generated based on common reactions, genes, and metabolites across the 1011 ssGEMs.

**Strain classification based on PCA, decision tree and cluster analysis.** The hierarchical cluster analysis based on the reaction existence in ssGEMs for yeast strains is based on R package––dendextend (https://CRAN.R-project.org/package = dendextend). For the PCA analysis of strains based gene (or reaction) existence in ssGEMs, R function-prcomp has been used in this article. The decision tree classification of strains according to the maximum growth rate on different carbon sources was carried out using the R package––rpart (https://cran.r-project.org/web/packages/rpart/). For the hyperparameters tuning, two R packages—ParamHelpers (https://CRAN.R-project.org/package = ParamHelpers) and mlr (https://CRAN.R-project.org/package=mlr) were further used.

**Protein structure collection for proYeast8^DB.** To establish the protein 3D structure models for all genes from yeast GEM (and a few metabolic genes not included in current Yeast8), all the protein structures of *S. cerevisiae* S288C from the SWISS-MODEL database[67] (https://Swissmodel.expasy.org) on 20 July 2018 were downloaded. The total number is about 20332 PDB files including the 8109 modelling homology PDB files (PDB_homo) and 12223 experimental PDB files (PDB_ex). Meanwhile all the PDB_ex of *S. cerevisiae* S288C stored in RCSB PDB[54] database were further downloaded. The protein sequences contained in each PDB_ex were also downloaded. The above two sources of PDB files were merged to obtain the comprehensive PDB files database for *S. cerevisiae* S288C. With the metabolic gene list of *S. cerevisiae* S288C to query PDB files database, most genes, with the exception of roughly 217 proteins (in Yeast8.3) could be found in the

related PDB files. To fill this gap, the SWISS-MODEL web service was further used to build the PDB_homo for 217 proteins. As a result, each of metabolic protein could have at least one PDB file. All the original proteins annotation, like the residues sequence and protein length, were downloaded from the SGD database.

Once the PDB files were collected, the parameters of PDBs were extracted and calculated for quality analysis. As for the PDB_homo, the default parameters from the ftp of the SWISS-MODEL database were obtained, and included the protein UniProt ID, the protein length, the related PDB ID (connected with chainID), the structure sources, the coordinates of proteins residues covered with PDB structures, the coverage, the resolution, and QMEAN. As for PDB_homo, besides the above default parameters from the SWISS-MODEL database, a greater number of parameters were obtained by parsing the PDB_homo atom files provided by the SWISS-MODEL with an in-house python script, which included the methods used to obtain the PDB files, the model template, the protein oliga state, the GMQE, QMN4, sequence identity (SID), and sequence similarity (SIM). In summary, each PDB_homo contains 18 parameters for further PDB quality analysis.

Some of PDB_ex parameters, like coverage and template ID can also be found from the SWISS-MODEL database. The other important parameters like resolution, ligands, and oliga state were obtained by parsing PDB_ex files from RCSB PDB database using (https://github.com/williamgilpin/pypdb). The chainID for each PDB_ex was downloaded from the SIFTS database[68].

**Quality analysis of protein 3D structure.** As one protein could be connected with several PDB files in different quality levels, it is essential to filter out the PDB of low quality. In this work, mainly four import parameters, that are sequence identity (SI), sequence similarity (SS), resolution, and QMEAN, were used to classify the PDB_homo. By using a simple normal distribution to describe all these parameters of PDB_homo, a Z score test can be done to calculate the threshold value for P value set at 0.1. The cut-off value of sequence identity, the sequence similarity, resolution, and QMEAN are 17.58, 0.25, 3.8 Å, and −6.98 respectively. As stated in the SWISS-MODEL database, however, a PDB_homo with the QMEAN smaller than −4 is of low quality. To ensure PDB_homo of higher quality in this work, the critical parameters are reset as the following: QMEAN ≥ −4, SI ≥ 0.25, SS ≥ 0.31, and Resolution ≤ 3.4 Å.

In order to check whether there exists a gap in the PDB_ex files, all residue sequences from PDB databases for each chain of one PDB file were downloaded. At some points, however, residue sequences provided by PDB databases were not consistent with residue sequences contained in the structure. To solve this issue, a Biopython package[69] was used to obtain residue sequences for each chain of one PDB file. Next, all residue sequences were blasted with original protein sequences for *S. cerevisiae* S288C from SGD with the aid of Diamond[66] in order to check whether existed gaps (mismatches or mutations) in the residue sequences from PDB_ex when compared with the original residue sequences. The PDB_ex has been chosen with the thresholds: pidentity = 100 and resolution ≤ 3.4 Å; otherwise a PDB_homo from SWISS-MODEL database will be used.

**Establishing relations of protein domain, gene, protein and reactions (dGRPs).** In this work, the Pfam32.0 database[70] (https://pfam.xfam.org/) was mainly used to annotate the domain information of proteins from *S. cerevisiae* S288C. If a structure covered all residues of any given domain, it was assigned to that very domain. For each domain, the coordinates of start and end, the name, the domain function description, the domain type, e_value, the related PDB ID, and protein ID, were all summarised. According to the GPRs of Yeast8, the relation between gene ID and reaction ID could be obtained. Following this, the domain information could be connected with each pair of gene and reaction based on the ID mapping.

**SNP collection and relative coordinates mapping.** Starting from the vcf file provided by the recent 1011 yeast strains genomes sequencing projects[40] the homozygous SNP from the massive data file (Supplementary Fig. 10a) were firstly extracted. The SNPs of low total quality with depth being <2.0, mapping quality <40, genotype quality < 30, and Genotype depth <5 were filtered out based on a series of standard parameters according to the Broad Institute Genome analysis Toolkit (GATK)[71].

After filtration, the reliable SNP can be obtained for each strain. The data furthermore contains each SNP's strain name, chromosome, coordinates, ref, and alt nucleotide base. In the annotation phase, the SNP type and related gene names were further annotated based on the coordinates and the annotation information of *S. cerevisiae S. cerevisiae* S288C reference genome (version R64-1-1) from NCBI. If the SNP was not located on CDS zone of gene, it was classified as a type of 'INTEGENIC'. If not this classification, it was otherwise given a gene systematic name, consistent with the gene name format in Yeast8. Based on the above SNP annotation information only those belonging to the metabolic genes (gene list in Yeast8 and some other metabolic genes not contained in Yeast8 until now) were chosen. According to the SNP annotation information and the protein sequences of the related genes, the SNPs are classified as the sSNP (synonymous single nucleotide polymorphism) and nsSNP (nonsynonymous single nucleotide polymorphism). The relative numbers of sSNPs and nsSNPs for each gene were

calculated, which is equal to the total sSNPs or nsSNPs divided by the related protein length.

Before mapping, the coordinates of mutated residues from each nsSNP need to be calculated. Firstly, the relative coordinates of mutated residues on the original protein sequence can be obtained based on the coordinates of nsSNP on the chromosome. Following this, according to the coordinates mapping between the original protein sequences and the relative residues coordinates in the proteins structure, the relative coordinates of the mutated residues in the protein structures can be estimated and used in the following calculation.

**CLUMPS method to calculate *p*-values of mutation enriched PDB files.** Referring to Kamburov's method[45], a WAP score to calculate the pairwise distances between mutated residues for a protein 3D structure.

$$WAP = \sum_{q,r} n_q n_r e^{-\frac{d_{q,r}^2}{2t^2}} \quad (7)$$

Where $d_{q,r}$ in this article is defined as the Euclidean distance (in Å) between α carbons of any two mutated residues. $t$ is defined as a 'soft' distance threshold, which equals to 6 Å. $n_q$ and $n_r$ are the normalised numbers of samples contains the mutations using the followed sigmoidal Hill function:

$$n_q = \frac{N_q^m}{\theta^m + N_q^m} \quad (8)$$

Where $Nq$ is the number of samples with a missense mutation impacting residue q of the protein and $\theta = 2$ and $m = 3$ are parameters of the Hill function controlling the critical point (centre) and steepness of the sigmoid function, respectively. Formula (2) was used to normalise the sample number contained in residue mutations q and r, both of which can avoid the impact of higher frequent mutated residues in the samples. A detailed description of each formula can be found in Kamburov's article[45].

The CLUMPS method can be divided into four steps. Firstly, prepare the needed SNP information and structure information of one protein. Secondly, with the normalised mutation number occurring in specific positions, calculate the WAP scores of the samples. Next, assuming that the uniform distribution of mutations across the protein residues covers the given structure, calculate each WAP score in 10 randomisations to obtain the null distribution. During the sampling process, the mutation number of residues occurring in random locations was kept the same as the original values. Lastly, calculate the right tailed $P$ value in the null distribution for the given mutated protein structures based on the original WAP score and all the sampled WAP scores. The right tailed $P$ value is defined as the number of samples with WAP scores larger than the original WAP scored, divided by the total number of samples.

For proteins with $P$ value smaller than 0.05 from strains group of "Bioethonal" and "Wine", GO-enrichment analysis using DAVID6.7 on-line web service[72] was carried out.

**Hotspot analysis of nsSNP mutation.** The hotspot analysis pipeline for yeast mainly refers to Niu et al.'s work[49]. All the SNP and structure information (similar to CLUMPS' analysis method) were prepared for a group of strains with specific phenotypes. Before carrying out the cluster analysis, the mutated paired residues of significance were filtered according to reference[49]. These important paired residues should meet the followed three criteria: the distance between two residues should be smaller than 10 Å for all the intramolecular clusters analysis; the two residues should be separated by at least 20 residues in the original protein sequence; and a permutation method ought to be used to calculate the $P$ value for each paired residues (Eq. 9), with a threshold set at 0.05.

$$P \text{ value} = \frac{n_1}{n_2} \quad (9)$$

Where $n_1$ is the number of paired residues with the distance smaller than that in the paired residues of target and $n_2$ is the total number of paired residues.

Once the paired residues of significance have been obtained, the clusters made up of paired residues were obtained based on the undirected graph theory, which was realised using the function 'decompose.graph' from the R package igraph (https://igraph.org/). For each cluster, its closeness can be calculated using the function of 'closeness.residual' from the R package entiserve[73]. The detailed principle could be also find in the original research[49]. As the last step, when a cluster was estimated, the $P$ value was calculated based on the CLUMPS analysis pipeline in this work.

**Prediction of mutations function.** To look into the effect of the mutations, mutfunc[48] (http://mutfunc.com/) and SIFT (http://snpeff.sourceforge.net/SnpSift.html) were employed to predict the potential effects of nsSNP on protein function.

**Growth test using Biolog with different substrate sources.** The Phenotype MicroArray (PM) system was used to test growth on every carbon, nitrogen, phosphorus and sulphur sources[74]. A total of 190 carbon sources, 95 nitrogen sources, 95 phosphorus, and sulphur sources were tested. The PM procedures for *S. cerevisiae* S288C were based on the protocol of Yeast version of the PM system.

**Growth profiling in different media.** A total of 14 carbon sources and 23 nitrogen sources were combined by orthogonal experiments. Every carbon source and nitrogen source used in the medium were the same C-mole and N-mole as glucose (20 g L$^{-1}$ glucose) and ammonium sulphate (7.5 g L$^{-1}$ (NH$_4$)$_2$SO$_4$), respectively. For all other substrate sources, the same minimal medium was used (14.4 g L$^{-1}$ KH$_2$PO$_4$, 0.5 g L$^{-1}$ MgSO$_4$·7H$_2$O, trace metal and vitamin solutions)[75]. Strains were cultivated in 96-well plates, and growth performance was determined with Growth Profiler 960 (Enzyscreen B.V., Heemstede, The Netherlands). The maximum specific growth rate ($\mu_{max}$) was calculated with the R package—growthrates (https://github.com/tpetzoldt/growthrates).

**Statistical analysis.** For two group comparison in this work, a two tailed Wilcoxon rank sum test was used.

**Reporting summary.** Further information on research design is available in the Nature Research Reporting Summary linked to this article.

## Data availability
The yeast-GEM project with all related data sources can be found at https://github.com/SysBioChalmers/yeast-GEM, and a full documentation on how the repository works and how to contribute is available in https://github.com/SysBioChalmers/yeast-GEM/blob/master/.github/CONTRIBUTING.md. The panYeast with all related data sources can be found in https://github.com/SysBioChalmers/panYeast-GEM. The genomic data used in this work is from[40].

## Code availability
Matlab scripts for development of yeast GEM and panYeast can be found in the above two corresponding repositories. R scripts to carry out the CLUMPS and hotspot analysis can be found at: https://github.com/SysBioChalmers/proYeast8-GEM.

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

## Acknowledgements

We would like to thank Pınar Kocabaş, Hao Wang, Jonathan Robinson, and Rui Pereira for their valuable input. This project has received funding from the Novo Nordisk Foundation (grant no. NNF10CC1016517), the Knut and Alice Wallenberg Foundation, and the European Union's Horizon 2020 research and innovation program with projects DD-DeCaF and CHASSY (grant agreements No 686070 and 720824). Open access funding provided by Chalmers University of Technology.

## Author contributions

H.L., F.L., B.J.S., E.J.K. and J.N. conceived the study. H.L., F.L., B.J.S., Z.Z. and E.J.K contributed to the yeast GEM update. B.J.S. and I.D. developed ecYeast8 and performed the corresponding simulations. H.L., F.L. and E.J.K. designed and generated panYeast8 and the 1,011 strain specific GEMs. H.L. designed and contributed to the completion of proYeast8$^{DB}$. G.L. assisted in the protein structure data collection for proYeast8$^{DB}$. Z.Z. and F.L. designed and performed the experiments. Z.Z. and H.L. designed and contributed to the yeast map. S.M., P.M.A., D.L., C.L., M.E.B., N.S. and E.J.K. provided support and ideas for yeast GEM development using version control. H.L., F.L. and J.N. wrote the original manuscript. All authors read, edited, and approved the final paper.

## Additional information

**Competing interests:** The authors declare no competing interests.

