## [Peer Review File · Nature Communications]

REVIEWERS' COMMENTS:

Reviewer #1 (Remarks to the Author):

The manuscript presents a new consensus metabolic model of *S. cerevisiae*. This is an important work with many novel contributions which will serve as an important resource and a knowledge reference for the broader biological community.

Specific comments:

1. The authors developed a new and updated version of the yeast genome scale model, yeast8. They have performed an excellent annotation and curation of the model using the best gene and reaction annotation databases. More interestingly, they also used Biolog experiments of growth on different carbon and nitrogen substrates and metabolomics analysis to improve the annotation, introduce new reactions, and update the (important) "biomass equation" with unprecedented quality and confidence. It is one of the very few examples that this has been done during GEM development by the same (or collaborating) group(s). The curation of the lipids is also one of the best and most comprehensive for yeast metabolism. I think that this procedure should serve as a paradigm and reference for future work in building GEMs.
2. The author integrated constraints for enzyme activity (enzyme amount and specific activity) and they validated the performance and importance of accounting for this constraints. This is the first time that this has been done for yeast and, most important, at this scale.
3. An impressive and very important contribution of this work is the construction, thorough curation, and comprehensive analysis of the panYeast and coreYeast. This is the first time that more than 1K yeast strains are analyzed through such systems biology and functional genomics pipeline. The ability of the model to capture and explain the variable yield of the strains on alternative substrates suggests to me that the panYeast model analysis is not a simple gene and reaction repository but a functional and useful tool and framework.
4. The SNP analysis in the context of the GEMs is very impressive. The authors have organized a very powerful and robust workflow and they have demonstrated how it can be used to assign systems function to SNPs. The results presented in the manuscript validate the procedure and demonstrate its potential. While one can ask for further analysis and other questions, I believe that any additional question relating SNPs to function is specific to a problem under study. The work here demonstrates that it can be done and how it should be done.

Overall, this is a very important work which will have a broad impact. The group has done excellent work in the development of the models and of the pipeline, they have used the state-of-the-art bioinformatics methods and databases, and they have demonstrated the value of their work with some studies that themselves are novel contributions to our knowledge of yeast genomics and physiology.

Reviewer #2 (Remarks to the Author):

The authors present their important work on establishing a platform for continuous community curation and expansion of genome-scale metabolic model of *Saccharomyces cerevisiae* while releasing the latest version of the model. In addition, an evolutionary-ecological aspect is taken as strain-specific models are generated for the recently sequenced 1011 *S. cerevisiae* strains (DOI: 10.1038/s41586-018-0030-5) and strain-dependent differences in *S. cerevisiae* metabolism uncovered. The authors further release extended versions of the latest *S. cerevisiae* models with enzyme *k_{cat}* values (turnover numbers) or protein structures introduced and elucidate the benefits of these additional features.

There are several *S. cerevisiae* genome-scale metabolic models used by the community currently as none of versions is found clearly better than the others (e.g. DOI: 10.1016/j.meteno.2016.05.002). The different versions are being independently curated and

used as starting points for models of other yeasts (DOI: 10.1093/femsyr/fox050). The reconstruction of the first consensus genome-scale metabolic model of *S. cerevisiae* was a community effort (DOI: 10.1038/nbt1492), but a platform for continuous community curation and expansion has been lacking. The establishment a platform for continuous community curation and expansion is very important for efficiently enriching the community knowledge into common knowledgebase.

While I consider the work very important and interesting, I have some concerns on how it is presented in the manuscript. Please, find my concerns detailed below.

There are different requirements for a genome-scale metabolic model if it is to act as a knowledgebase or as a model with high predictive power for metabolic phenotypes. A knowledgebase should ideally include all, even minor activity, metabolic reactions and include gene annotation on a reaction even when the the gene encodes an enzyme whose minor side-activity the particular reaction is. However, in a stoichiometric modelling framework relying on linear programming on the prediction, the predictive power is reduced if minor activity reactions are included without constraints on them. A discussion on these very different purposes of genome-scale metabolic models and, considering the different purposes, how the community should build the model(s) forward is missing. ecYeast8 does not fully solve this when kcat values are not available extensively enough or protein content constraints difficult to set under particular conditions.

In the paragraph, starting on line 52, it is essential to clarify that introducing enzyme kinetics into genome-scale metabolic refers here to enzyme kcat values and not considering the metabolite concentration effects on fluxes. Later, a term "enzyme constraints" is used which is more appropriate. Further, clarification is also needed to specify that the environmental conditions referred to in this paragraph are other than nutritional conditions. Metabolic phenotype predictions under different nutritional conditions are standard simulations of genome-scale metabolic models. Finally, it remains unclear what is meant by "advanced mutation mapping" here. Please, revise this important part of introduction to clarify the value and impact of introducing kcat's and protein structures to *S. cerevisiae* models.

On line 106 SBML model format is referred to as the format for metabolic maps to be visualized. Is this really correct or was the aim to refer to SBGN graphical format? In addition, the maps are not found with the given link on line 561. Please, provide a correct link, and correct the map format.

On line 114 chemostat and batch conditions are referred to which may not be clear for readers with other than industrial biotechnology background. I would suggest using glucose-limited and glucose-excess conditions instead.

On lines 147 and 148 "found" is repeated, please, revise.

In the paragraph starting on line 171 simulated maximum amino acid yields are discussed. Amino acid biosynthesis pathways are well-conserved. If the different strains of *S. cerevisiae* are to show different theoretical maximum amino acid yields, a discussion is expected on the underlying pathway differences. Please, clarify the underlying pathway differences.

On line 306 the authors refer to finding metabolic differences between the *S. cerevisiae* strains, "several of which are related to evolutionary adaptation". It is unclear how in this study the metabolic differences could be concluded to be related to evolutionary adaptation. Please, clarify.

Line 491 refers to "metabolites contained in new GPRs". However, GPRs are gene-protein-reaction rules of genetic underpinnings of reactions. How are metabolites involved here? Please, clarify.

On line 548, please clarify that glucose and oxygen uptake fluxes are negative and therefore the

lower bounds are fixed to represent the maximum uptakes.

On lines 722-723 parameters for SNP filtration are given. The parameters are very loose particularly for the total quality by depth and genotype quality. Please, justify the choices as the SNP set is expected to contain a lot of false positive SNPs.

Figure 3, the number of strains belonging to the classes of different ecological origin differ a lot. Therefore it is essential to plot in subfigures a, b, and c all the points visible. What kinds of substrates does the subfigure c visualize (e.g. carbon sources)? Please, clarify.

REVIEWERS' COMMENTS:

Reviewer #1 (Remarks to the Author):

The manuscript presents a new consensus metabolic model of *S. cerevisiae*. This is an important work with many novel contributions which will serve as an important resource and a knowledge reference for the broader biological community.

Specific comments:

1. The authors developed a new and updated version of the yeast genome scale model, yeast8. They have performed an excellent annotation and curation of the model using the best gene and reaction annotation databases. More interestingly, they also used Biolog experiments of growth on different carbon and nitrogen substrates and metabolomics analysis to improve the annotation, introduce new reactions, and update the (important) "biomass equation" with unprecedented quality and confidence. It is one of the very few examples that this has been done during GEM development by the same (or collaborating) group(s). The curation of the lipids is also one of the best and most comprehensive for yeast metabolism. I think that this procedure should serve as a paradigm and reference for future work in building GEMs.
2. The author integrated constraints for enzyme activity (enzyme amount and specific activity) and they validated the performance and importance of accounting for this constraints. This is the first time that this has been done for yeast and, most important, at this scale.
3. An impressive and very important contribution of this work is the construction, thorough curation, and comprehensive analysis of the panYeast and coreYeast. This is the first time that more than 1K yeast strains are analyzed through such systems biology and functional genomics pipeline. The ability of the model to capture and explain the variable yield of the strains on alternative substrates suggests to me that the panYeast model analysis is not a simple gene and reaction repository but a functional and useful tool and framework.
4. The SNP analysis in the context of the GEMs is very impressive. The authors have organized a very powerful and robust workflow and they have demonstrated how it can be used to assign systems function to SNPs. The results presented in the manuscript validate the procedure and demonstrate its potential. While one can ask for further analysis and other questions, I believe that any additional question relating SNPs to function is specific to a problem under study. The work here demonstrates that it can be done and how it should be

done.

Overall, this is a very important work which will have a broad impact. The group has done excellent work in the development of the models and of the pipeline, they have used the state-of-the-art bioinformatics methods and databases, and they have demonstrated the value of their work with some studies that themselves are novel contributions to our knowledge of yeast genomics and physiology.

Response:

We thank the reviewer for the kind comments and stating the importance of our work.

Reviewer #2 (Remarks to the Author):

The authors present their important work on establishing a platform for continuous community curation and expansion of genome-scale metabolic model of *Saccharomyces cerevisiae* while releasing the latest version of the model. In addition, an evolutionary-ecological aspect is taken as strain-specific models are generated for the recently sequenced 1011 *S. cerevisiae* strains (DOI: 10.1038/s41586-018-0030-5) and strain-dependent differences in *S. cerevisiae* metabolism uncovered. The authors further release extended versions of the latest *S. cerevisiae* models with enzyme *k_{cat}* values (turnover numbers) or protein structures introduced and elucidate the benefits of these additional features. There are several *S. cerevisiae* genome-scale metabolic models used by the community currently as none of versions is found clearly better than the others (e.g. DOI:10.1016/j.meteno.2016.05.002). The different versions are being independently curated and used as starting points for models of other yeasts (DOI:10.1093/femsyr/fox050). The reconstruction of the first consensus genome-scale metabolic model of *S. cerevisiae* was a community effort (DOI: 10.1038/nbt1492), but a platform for continuous community curation and expansion has been lacking. The establishment a platform for continuous community curation and expansion is very important for efficiently enriching the community knowledge into common knowledgebase.

While I consider the work very important and interesting, I have some concerns on how it is presented in the manuscript. Please, find my concerns detailed below.

Response:

We thank the reviewer for the kind comments, and we have refined the manuscript based on the excellent questions and suggestions. All revised contents were marked as red in manuscript.

There are different requirements for a genome-scale metabolic model if it is to act as a knowledgebase or as a model with high predictive power for metabolic phenotypes. A knowledgebase should ideally include all, even minor activity, metabolic reactions and include gene annotation on a reaction even when the gene encodes an enzyme whose minor side-activity the particular reaction is. However, in a stoichiometric modelling framework relying on linear programming on the prediction, the predictive power is reduced if minor activity reactions are included without constraints on them. A discussion on these very different purposes of genome-scale metabolic models and, considering the different purposes, how the community should build the model(s) forward is missing. ecYeast8 does not fully solve this when *kcat* values are not available extensively enough or protein content constraints difficult to set under particular conditions.

Response:

We thank the reviewer for this excellent comment. Yeast8 is primarily a constraint-based model and can predict the common phenotypes of yeast. On the other hand, it can be regarded a strain specific knowledge database as new metabolic knowledge of *S. cerevisiae* will be merged continuously into Yeast8.

However as the growth conditions or gene background differs, a condition specific GEM becomes more useful and important as it could have a better prediction performance compared with the general GEM¹. In our study we verified that condition specific models extracted from the generic GEM according to reaction existing scores based on transcriptomics or proteomics evidences have better predictive power². We do, however, believe that Yeast8 is also a reconstruction as all the source data updated in the model is of high quality and can be tracked carefully using Git and GitHub so that everyone could check the quality of model curation if new information is added. This is setting new standards for building both reconstructions and simulation ready models. It has also recently been suggest that the annotation and model quality control analysis should be done in a standard way as reported in a recent paper³.

We have added a comment about this in our revised manuscript.

In the paragraph, starting on line 52, it is essential to clarify that introducing enzyme kinetics

into genome-scale metabolic refers here to enzyme k_{cat} values and not considering the metabolite concentration effects on fluxes. Later, a term “enzyme constraints” is used which is more appropriate. Further, clarification is also needed to specify that the environmental conditions referred to in this paragraph are other than nutritional conditions. Metabolic phenotype predictions under different nutritional conditions are standard simulations of genome-scale metabolic models. Finally, it remains unclear what is meant by “advanced mutation mapping” here. Please, revise this important part of introduction to clarify the value and impact of introducing k_{cat} ’s and protein structures to *S. cerevisiae* models.

Response:

Very good suggestions. The “enzyme kinetics” was changed into “enzyme constraints”. The “environmental conditions” was changed into “environmental conditions other than nutritional conditions”. Also, we added the followed contents to display the value and impact of introducing k_{cat} ’s and protein structures to metabolic models.

“The GEMs constrained with k_{cat} and enzyme abundances could directly integrate the proteomics data and correctly predict the cellular phenotype under conditions of stress⁴. The GEMs with protein 3D structures connect the structures-related parameters and genetic variation⁵ with cellular metabolism⁶, thus enlarging the prediction scope of GEMs.”

On line 106 SBML model format is referred to as the format for metabolic maps to be visualized. Is this really correct or was the aim to refer to SBGN graphical format? In addition, the maps are not found with the given link on line 561. Please, provide a correct link, and correct the map format.

Response:

Here the SBML model format is aimed to refer to SBGN graphical format. We have rephased the sentences. Also a new link was provided for the yeast whole map, which can be downloaded from <https://github.com/SysBioChalmers/Yeast-maps/blob/master/SBMLfiles/Yeast8.xml>.

On line 114 chemostat and batch conditions are referred to which may not be clear for readers with other than industrial biotechnology background. I would suggest using glucose-limited and glucose-excess conditions instead.

Response:

We have corrected it.

On lines 147 and 148 “found” is repeated, please, revise.

Response:

We have corrected it.

In the paragraph starting on line 171 simulated maximum amino acid yields are discussed. Amino acid biosynthesis pathways are well-conserved. If the different strains of *S. cerevisiae* are to show different theoretical maximum amino acid yields, a discussion is expected on the underlying pathway differences. Please, clarify the underlying pathway differences.

Response:

We agree with the reviewer that amino acid synthesis pathways are well conserved among these 1011 yeast stains. The difference in theoretical maximum amino acid yields are due to two main reasons. The first one is that some strains with low maximal amino acids yields use fermentation instead of respiration to produce energy for amino acid synthesis, because they don't have all essential subunits for respiration complexes. Those strains should not be able to use non-fermentable carbon sources such as ethanol and glycerol for growth as phenotypes if the respiration is impaired as our model indicated. According to our models, *S. cerevisiae* AAH uses fermentation to produce energy for amino acid synthesis and has low maximal amino acid yields. This strain grows poorly on the medium with ethanol (relative growth rate: 0.00562701) and glycerol (relative growth rate: 0.00617284) as the main carbon sources according to literature⁷.

Secondly, a few important synthesis reactions for some amino acids are missing in several strains, resulting in non-production of the corresponding amino acids. For example, in *S. cerevisiae* BLT and AHG, YHR208W or its ortholog 179-augustus_masked-2806-CPI_4 (valine transaminase, mitochondrial) in the last step of valine synthesis are missing, so the maximal yield of valine for those strains decrease to zero. Similarly, YNL220W (adenylosuccinate synthase) related for histidine synthesis is missing in strain *S. cerevisiae* ABM, so the related maximal histidine yield is decreased to zero in our simulation. There is only one strain *S. cerevisiae* SACE_GAV, which does not have the gene YDR007W (phosphoribosylanthranilate isomerase) in the third step of tryptophan biosynthesis, so the related maximal yield for tryptophan decreases to zero. These results together may indicate the auxotrophic phenotypes existing in part of these 1011 strains.

In summary, due to the differences in the energy pathway for the ATP generation and the absence of some essential genes in amino acid synthesis, the predicted maximal yield of

amino acids varies among 1011 strains, illustrating that the simulation with strain specific models could help to explore the relations between the genotype and the phenotype. We clarify the underlying pathway differences in manuscript and add the detailed analysis in the Supplementary note 1.

On line 306 the authors refer to finding metabolic differences between the *S. cerevisiae* strains, “several of which are related to evolutionary adaptation”. It is unclear how in this study the metabolic differences could be concluded to be related to evolutionary adaptation. Please, clarify.

Response:

We appreciate the reviewer’s emphasis on the relation between the metabolic differences and the evolutionary adaptation. we can indeed not conclude evolutionary adaptation from this study, but it does give clues on the role of evolutionary adaptation. To be more accurate, we changed our statement to :

“However, through strain specific GEM simulations, we found subtle metabolic differences among the strains in the utilization of substrates and the maximum yield of 26 chemicals. Exploring these differences constrained with more physiological data can guide future metabolic engineering and help to evaluate the potential of any given strain for any desired product, as well as providing clues about the mechanisms of evolutionary adaptation.”

Line 491 refers to “metabolites contained in new GPRs”. However, GPRs are gene-protein-reaction rules of genetic underpinnings of reactions. How are metabolites involved here? Please, clarify.

Response:

We apologize for this mistake and it should be “metabolites contained in the newly added reactions”. We have corrected this in the revised version of the paper.

On line 548, please clarify that glucose and oxygen uptake fluxes are negative and therefore the lower bounds are fixed to represent the maximum uptakes.

Response:

We thank the reviewer for this comment. We have added the description in the revised version.

On lines 722-723 parameters for SNP filtration are given. The parameters are very loose particularly for the total quality by depth and genotype quality. Please, justify the choices as

the SNP set is expected to contain a lot of false positive SNPs.

Response:

We thank the reviewer for pointing out the parameter setting in the SNP filtration. To be more accurate, it should be "SNPs of low total quality with depth (QD) being < 2.0, mapping quality (MQ) < 40, genotype quality (GQ) <30, and Genotype depth (DP) <5 were filtered out based on a series of standard parameters according to the Broad Institute Genome analysis Toolkit (GATK)". In fact, these parameters are widely used in SNP filtration and an example could be found in Gallone's paper published in Nature⁸. Description of QD and MQ can also be found from the GATK tutorial

(<https://software.broadinstitute.org/gatk/documentation/article.php?id=2806>) while description of the GQ setting can be found in⁹.

While it should be noted that the mutation enrichment analysis was conducted for the whole protein 3D structure or specific zones from a whole structure. Thus, the significant mutation clusters (or hotspots) are always made up of by multiple mutations from different sites, which, to some extent, could reduce the analysis of bias due to the presence of false positive SNPs. In our work, the meaningful proteins targets related to cellular phenotypes could be identified using the CLUMPS and hotspot analysis pipeline, which also justifies the present parameters setting.

Figure 3, the number of strains belonging to the classes of different ecological origin differ a lot. Therefore it is essential to plot in subfigures a, b, and c all the points visible. What kinds of substrates does the subfigure c visualize (e.g. carbon sources)? Please, clarify.

Response:

We thank the reviewer for the nice suggestion! Now in the revised Fig.3a, 3b & 3c, all points are displayed in the figure (as an example, you can find the comparison between the new Fig.3a and the original Fig.3a below). It should be noted that in one ecological origin, possibly, a lot of stains have the same value in the number of reactions and utilized substrates, as well as in the yield of biomass so the points in each column can't represent the real number of strains contained from each ecological origin. We added an additional figure in the new Fig.3a to represent the strain numbers from each ecological origin. The new Figure could help to understand how the data points were distributed among the different ecological origins.

Original Fig.3a

New Fig.3a

In Fig.3c, the substrates include 58 carbon sources, 46 nitrogen sources, 41 phosphate sources and 12 sulphate sources. It should be noted that for *in silico* carbon sources utilization, ammonium is used as the sole nitrogen source; while for nitrogen sources, glucose is used as the sole carbon source and for phosphate and sulphate sources, ammonium and glucose are used in the minimal media for yeast cell growth. We have added the substrates description in the legend of Fig.3

Reference

1. Monk, J.M. et al. iML1515, a knowledgebase that computes Escherichia coli traits. *Nat Biotechnol* **35**, 904-908 (2017).
2. Mardinoglu, A. et al. Integration of clinical data with a genome-scale metabolic model of the human adipocyte. *Mol Syst Biol* **9**, 649 (2013).
3. Lieven, C. et al. Memote: A community driven effort towards a standardized genome-scale metabolic model test suite. *bioRxiv*, 350991 (2018).
4. Sanchez, B.J. et al. Improving the phenotype predictions of a yeast genome-scale metabolic model by incorporating enzymatic constraints. *Mol Syst Biol* **13**, 935 (2017).

5. Chang, R.L. et al. Structural systems biology evaluation of metabolic thermotolerance in *Escherichia coli*. *Science* **340**, 1220-1223 (2013).
6. Brunk, E. et al. Recon3D enables a three-dimensional view of gene variation in human metabolism. *Nat Biotechnol* **36**, 272-281 (2018).
7. Peter, J. et al. Genome evolution across 1,011 *Saccharomyces cerevisiae* isolates. *Nature* **556**, 339-344 (2018).
8. Gallone, B. et al. Domestication and Divergence of *Saccharomyces cerevisiae* Beer Yeasts. *Cell* **166**, 1397-1410 e1316 (2016).
9. Lazaridis, I. et al. Ancient human genomes suggest three ancestral populations for present-day Europeans. *Nature* **513**, 409-413 (2014).